Evaluating deep learning variants for cyber-attacks detection and multi-class classification in IoT networks

http://orcid.org/0009-0001-0117-4390 Abbas Sidra 1 sidraabbas@ieee.org
Bouazzi Imen 2
Ojo Stephen 3
Al Hejaili Abdullah 4
Sampedro Gabriel Avelino 5 6
Almadhor Ahmad 7
http://orcid.org/0000-0001-6207-1347 Gregus Michal 8 michal.gregus3@uniba.sk
1 Department of Computer Science, COMSATS Institute of Information Technology , Islamabad , Pakistan
2 Department of Industrial Engineering, King Khalid University , Abha , Saudi Arabia
3 Department of Electrical and Computer Engineering, College of Engineering , Anderson University, Anderson, SC , United States of America
4 Computer Science Department, University of Tabuk , Tabuk , Saudi Arabia
5 Faculty of Information and Communication Studies, University of the Philippines Open University , Los Baños , Philippines
6 Center for Computational Imaging and Visual Innovations, De La Salle University , Manila , Philippines
7 Department of Computer Engineering and Networks, College of Computer and Information Sciences, Al Jouf University , Sakaka , Saudi Arabia
8 Information Systems Department, Faculty of Management, Comenius University in Bratislava , Bratislava , Slovak Republic
Alarcon-Aquino Vicente
Electronic publication date: 2024 Jan 16
Publication date: 2024
Volume: 10
Electronic Location ID: e1793
Received 2023 Sep 20; Accepted 2023 Dec 13
Copyright: © 2024 Abbas et al.
Copyright year: 2024
Copyright holder: Abbas et al.
License: This is an open access article distributed under the terms of the Creative Commons Attribution License, which permits unrestricted use, distribution, reproduction and adaptation in any medium and for any purpose provided that it is properly attributed. For attribution, the original author(s), title, publication source (PeerJ Computer Science) and either DOI or URL of the article must be cited.
License URL: https://creativecommons.org/licenses/by/4.0/

Keywords: Cyber-attacks, IoT, DDoS attacks, Deep learning

Funding: King Khalid University RGP2/470/44 This work was supported by the Deanship of Scientific Research at King Khalid University through large group Research Project under grant number RGP2/470/44. The funder had a role in the conceptualization, study design and decision to publish and preparation of the manuscript. The funder had no role in data collection and analysis.

==============================
The Internet of Things (IoT), considered an intriguing technology with substantial potential for tackling many societal concerns, has been developing into a significant component of the future. The foundation of IoT is the capacity to manipulate and track material objects over the Internet. The IoT network infrastructure is more vulnerable to attackers/hackers as additional features are accessible online. The complexity of cyberattacks has grown to pose a bigger threat to public and private sector organizations. They undermine Internet businesses, tarnish company branding, and restrict access to data and amenities. Enterprises and academics are contemplating using machine learning (ML) and deep learning (DL) for cyberattack avoidance because ML and DL show immense potential in several domains. Several DL teachings are implemented to extract various patterns from many annotated datasets. DL can be a helpful tool for detecting cyberattacks. Early network data segregation and detection thus become more essential than ever for mitigating cyberattacks. Numerous deep-learning model variants, including deep neural networks (DNNs), convolutional neural networks (CNNs), and recurrent neural networks (RNNs), are implemented in the study to detect cyberattacks on an assortment of network traffic streams. The Canadian Institute for Cybersecurity’s CICDIoT2023 dataset is utilized to test the efficacy of the proposed approach. The proposed method includes data preprocessing, robust scalar and label encoding techniques for categorical variables, and model prediction using deep learning models. The experimental results demonstrate that the RNN model achieved the highest accuracy of 96.56%. The test results indicate that the proposed approach is efficient compared to other methods for identifying cyberattacks in a realistic IoT environment.

Introduction

The rapid advancement of connected technologies has contributed to deploying many Internet of Things (IoT) devices in numerous applications (Javed et al., 2022). Simultaneously, the issue of cyberattacks has become more challenging to address. Because most IoT gadgets have extremely low resources (i.e., computing and storage capability), they cannot implement sophisticated intrusion detection systems. Global smartphone data stream reportedly climbed by roughly 71% in 2017 compared to the previous year, and by 2022, nearly all cellular traffic data is expected to come from smart gadgets (Cisco, 2019). Compared to 2013, there was a roughly sevenfold rise in the amount of IoT viruses in 2017. Many Mirai-infected IoT gadgets concurrently launched a cyberattack with a peak bandwidth of 620 Gbps (Spamhaus Malware Labs, 2020). In 2018, 10,263 distinct IoT gadgets hosting botnets were found (Haider et al., 2016). The year 2017 witnessed the discovery of “IoTroop,” a further distributed denial of service (DDoS) assault originating from exploited IoT devices. During this significant assault that targeted the financial sectors, 13,000 IoT gadgets were utilized.

Today’s sophisticated and well-equipped cyber-criminals can successfully assault organizations like governments and enterprises (Mustafa et al., 2020). Currently, cybercrime is a massive industry with massive amounts of stolen data. Malware can be divided into numerous groups (Behal & Kumar, 2017). This entails a significant danger for all governments, corporations, and customers worldwide. We do not have to think back very far to recall the big attack on a Bangladeshi bank, during which an estimated USD 81 million was stolen. The fact that the bank’s systems were utilized to move huge quantities of money is an ongoing indicator of how successful such attacks can be. No business, regardless of size, is secure. According to statistics, 20% of impacted companies tumble into the compact business class, 33% into the SME class, and 41% into the big corporation class. The greater the hazard, the more crucial it is to be informed of the problems and safeguard sensitive data. Eighty-two percent of businesses have experienced at least three data-stealing attempts that have rendered their customers’ services useless. The organizations impacted by DDoS assaults outlined a 26% decline in operational efficiency and a 41% disruption of the targeted services (Behal & Kumar, 2017).

The attacks on malware networks, such as Bashlite, Mirai, Hajime, and others, provide the biggest challenge to IoT security. Network attacks include DDoS, pushing, identity theft, data leaking, and keylogging (Amini, Araghizadeh & Azmi, 2015). The botmasters perform distant network mapping for Operating System fingerprinting (OS fingerprinting), information collecting, and port scanning to locate their vulnerabilities and compromise these gadgets. Botnets launched most DDoS assaults to disturb services and blocked reputable users from using them (Koroniotis et al., 2019). DDoS attacks are possible in the IoT setting at the network and application layers (Hassija et al., 2019). Stealing data and confidentiality issues are distinct security risks in the IoT ecosystem’s application layer (Hassija et al., 2019). The study (Amin, Ahmad & Sang Choi, 2019) examined the difficulties for users of data exposure and privacy infringement.

Malicious individuals frequently enhance their skills, alter their strategies, and use cutting-edge technologies to perpetrate DDoS attacks. DDoS attacks can be detected, prevented, or mitigated using various techniques, but nasty people constantly develop novel ways to get around existing defenses (Behal & Kumar, 2017). One of the major risks to the network nowadays is DDoS activity. Recent DDoS attacks at the application layer of web servers have expanded widely, costing targets a tonne of money (Jiang et al., 2017). Attacks on the TCP/IP layer cripple online servers and have a cap on the number of requests that can be made per second. This category includes DDoS attacks, Slowloris, and zero-day attacks that exploit Apache or Windows flaws (Yusof, Ali & Darus, 2018). The strategies proposed to understand DDoS assaults at the TCP/IP layer are merely a part of DDoS occurrences at the application layer. The formulas for the decisions that detect application-layer attacks are extremely complex. The lack of infrastructure to detect such attacks is a particular class of jobs in identifying a DDoS epidemic at the TCP/IP layer (Yadav & Subramanian, 2016). DDoS attacks are difficult to identify since they are quite diverse and often occur with other cyberattacks.

The most effective way to prevent DDoS attacks is to take action immediately as they happen. The popularity of cyberattacks on internet-connected equipment has grown due to increased internet usage. Since ML and DL (Khuphiran et al., 2018), have started recognizing their enormous potential in various fields, industry and academia are exploring integrating ML and DL for cyberattack mitigation. Traditional approaches for risk identification need to be more accurate and respond gradually. Threats can be identified more effectively and precisely using ML techniques like random forest (Farnaaz & Jabbar, 2016), KNN (Li & Guo, 2007), and naive Bayesian (Panda & Patra, 2007). The subject matter specialist must select the characteristics for classification in machine learning. Within the deep learning model, feature selection takes place. A series of nonlinear types of layers are used by DL techniques like ANN (Li & Zhang, 2019), DNN, and RNN to learn different patterns from several labeled data. Therefore, DL can serve as a useful tool for cyberattack detection. After thoroughly evaluating the possibilities, we decided to use the DNN, CNN, and RNN Deep Learning frameworks in our experiment to detect cyberattacks. Experimental results show that it serves the intended purpose admirably.

Contribution

This research aimed to identify cyberattacks in IoT environments using deep-learning models on the CICDIoT2023 dataset. The research’s main contribution is given as follows. This research proposed multiple variants of deep learning models to detect cyberattacks in realistic IoT environments on various samples of network stream packets.

This study contributes to implementing an IoT cyberattacks dataset from CICIoT Dataset 2023, an actual dataset and standard for massive attacks in the Internet of Things environment. This research utilizes 47 features and 33 attacks for assessment.

This research uses robust scalar and label encoding techniques to preprocess the data and compare variants of deep learning models (DNNs, CNNs and RNNs).

The research evaluates the performance of the cyberattack detection system using different evaluation metrics. The outcomes show that the proposed approach is effective in accurately predicting cyberattacks.

Organization

The article’s organization is as follows: “Related Work” contains a study on relevant ML and DL techniques for cyberattack detection. “Proposed Methodology” describes the proposed approach that uses the CIC IoT dataset, data preparation, and deep learning models. “Results and Discussion” explains and discusses the findings. “Conclusion and Future Scope” contains the study’s conclusion and recommendations for further investigation.

Related work

Since the growing need for IoT automated network systems, IoT model complexity is increasing regularly. Because the gadgets broadcast data using wireless technology, they are considered easy targets. Every day, thousands of attacks surface due to the inclusion of new IoT protocols, which frequently cause the computing process to become worse, more unstable, and ineffective. The typical communication assault on a local network is limited to local nodes or tiny local domains. The assault found in IoT devices, however, spreads to a wide region and has disastrous effects. IoT security is harder and more intricate due to the heterogeneity and spread of IoT applications and service providers (Babu & Veena, 2021). In a study (Almaraz-Rivera, Perez-Diaz & Cantoral-Ceballos, 2022), the author builds a unique IDS based on ML and DL models, tackling the class imbalance issue in the Bot-IoT dataset. Researchers employed three alternative assortments of features for binary and multi-class categories to assess how the timings of the observations affect the predicts, which allowed the elimination of feature connections, which the Argus flow information encoder provided, while still attaining a median accuracy of >99%. According to the stacking generalization principle, the study (Dutta et al., 2020) proposes an ensemble technique that uses deep models and a meta-classifier. The method employs a two-step procedure for detecting network irregularities to increase the capabilities of the proposed methodology.

In Shurman, Khrais & Yateem (2020), the author proposed two methods for identifying DDoS assaults in IoT systems. The first approach detects IoT-DoS attacks using an IDS. The second approach uses DL models, namely LSTM networks built with the most recent dataset for DDoS assaults of this sort. Our test findings show that the proposed approaches can identify abnormal behavior, protecting the Internet of Things network from DoS and DDoS assaults. The author of study (Saba et al., 2022) offers a CNN-based approach for an intrusion-based IDS system, which effectively uses the IoT’s potential and allows users to investigate all IoT traffic effectively. The model’s accuracy was 99.51% during training and 92.85% during testing. A DDoS detection strategy based on ML approaches is proposed in an article (Seifousadati, Ghasemshirazi & Fathian, 2021) using the CICDDoS2019. It tested the most well-liked ML techniques and identified attributes most connected with projected classes. It was found that XGBoost and AdaBoost performed exceptionally well, accurately predicting internet traffic with an accuracy rate of 100%. The study (Alzahrani & Alzahrani, 2021) uses the CICDDoS2019 datasets to analyze the effectiveness of detection for DDoS assaults by implementing various ML methods in WEKA software. Using the random forest (RF) and decision tree (DT) methods produced the highest accuracy results in the assessment at 99%. However, it performs better because the DT computes in 4.53 s compared to 84.2 s for the RF.

In research (Abu Al-Haija & Zein-Sabatto, 2020), the authors present the thorough creation of a novel smart and automated deep-learning-based identification and categorization model for cyberattacks in IoT communication systems that use CNN capabilities. The simulation results showed that the proposed system had a superior 99.3% and 98.2% accuracy in classifying cyber-attacks. The study’s author, Pei, Chen & Ji (2019), proposes a machine learning-based DDoS assault identification process for extracting features and model assessment. Evaluating the data packages classified according to criteria enables the feature extraction stage to extract the DDoS attack traffic features with a high percentage. The machine learning model identification step uses the retrieved features as input characteristics, and the assault prediction model is developed using the random forest method. The trials’ findings show that the machine learning-based DDoS assault detection method effectively detects current DDoS assaults at a rate compatible with its detection rate.

In a study (Banitalebi Dehkordi, Soltanaghaei & Boroujeni, 2021), the author proposed a novel approach for identifying DDoS assault. The findings indicate that this method exceeds its competitors’ accuracy, attaining 99% for identifying DDoS assault in SDN. In Saghezchi et al. (2022), the author used machine learning (ML) to build various statistical models for broadband anomaly recognition and DDoS assault detection. The research uses network traffic information gathered from a real-world semiconductor manufacturing facility. The findings demonstrate that supervised techniques outperform unsupervised and semi-supervised counterparts regarding detection efficiency. Specifically, the decision tree model limits the false positive rate to 0.001 and achieves an accuracy of 0.999. In a study (Chen et al., 2020), the author proposes an IoT DDoS attack surveillance system with multiple layers, including IoT gadgets, pathways, SDN toggles, and cloud servers. The authors first constructed eight smart poles on the campus with various sensors to acquire sensor data for datasets. The authors then extract the properties depending on different forms of DDoS attacks. According to the experimental findings, DDoS attacks can be precisely detected through the multi-layer DDoS surveillance system.

In Kumari & Mrunalini (2022), a computational model for DDoS assault is presented. ML algorithms like LR and NB are implemented to identify attacks and typical situations. The pilot investigation utilizes the CAIDA 2007 Dataset. This study implements the Weka data mining platform, and the outputs of that platform are contrasted and assessed. In a study (Kumar et al., 2023), a framework based on “LSTM” is developed to identify DDoS assault in an instance of transmitted network streams. In the present work, the proposed system attained an accuracy of up to 98%. This study (Elsaeidy, Jamalipour & Munasinghe, 2021) develops a hybrid DL approach to identify DDoS and replication assaults on an actual intelligent city system. The effectiveness of the proposed hybrid model is assessed using realistic DDoS and replication assaults on real-world intelligent city datasets. For the environmental dataset (98.37%), intelligent river dataset (98.13%), and intelligent soil dataset (99.51%), the proposed model demonstrated excellent prediction rates. The study (Anwer et al., 2021) suggests a paradigm for identifying illegal internet traffic. The framework achieves much higher accuracy (85.34%) using ML classification-based techniques for harmful network traffic identification. The proposed framework was applied to the NSL KDD dataset.

Subsequently, several methods for detecting cyberattacks utilizing ML and DL techniques are presented in Table 1. However, they are constrained in their ability to perform more effectively due to their unwillingness to select characteristics and extraction techniques. This work provided an effective method for identifying cyberattacks using several deep-learning model variants.

Table 1 Summary of existing related work.

Reference	Focus	Technique	Limitation	
Neto et al. (2023)	Cyberattacks detection	RF	Low performance	
Kumar et al. (2023)	Cyberattacks detection	LSTM	Low performance	
Elsaeidy, Jamalipour & Munasinghe (2021)	Cyberattacks detection	Hybrid ML model	Low performance	
Anwer et al. (2021)	Cyberattacks detection	ML model	Low performance	
Abu Al-Haija & Zein-Sabatto (2020)	Cyberattacks detection	CNN	Low performance	

Proposed methodology

The proposed approach entails several steps, such as dataset collection from the CIC repository, data pre-processing, and three model building to predict cyberattacks. Figure 1 depicts the overview of the proposed architecture.

Figure 1 Proposed architecture overview.

Experimental dataset

This study used the CICIoT2023 Dataset, an actual dataset and benchmark for massive attacks in the Internet of Things environment. There are two distinct varieties of files for the CICIoT2023 dataset: pcap and csv. The initial data generated and gathered in various scenarios inside the CIC IoT network is in pcap files. All sent packets are included in these files, which can be used to design additional functionality. A fixed-size packet window summarizes the features taken from the original pcap files and is added to those files. A series of packets delivering data between two hosts is used to extract the features (Neto et al., 2023).

This research utilizes the 47 features for assessment. Using the retrieved features, we aggregate the values recorded in intervals size of 100 (Mirai UDPPlain, Mirai Greeth Flood, Mirai GREIP Flood, DoS UDP Flood, DoS TCP Flood, DDoS UDP Flood, DoS HTTP Flood, DDoS TCP Flood, DoS SYN Flood, DDoS UDP Fragmentation, DDoS SYN Flood, DDoS HTTP Flood, DDoS SynonymousIP Flood, DDoS RSTFIN Flood, DDoS ICMP Fragmentation, DDoS ICMP Flood, DDoS SlowLoris, DDoS ACK Fragmentation and DDoS PSHACK Flood) and 10 (Dictionary brute force, Ping Sweep, Vulnerability Scan, Host Discovery, Backdoor Malware, Command Injection, OS Scan, SQL Injection, DNS spoofing, XSS, Browser Hijacking, Uploading Attack, MITM ARP spoofing, Benign Traffic and Port Scan) packages to reduce fluctuating data sizes (such as Command_Injection and DDoS). Subsequently, the csv file dataset is used for this research. The generated csv datasets indicate the features of every data block. Additionally, each attack used in this research has unique properties. For instance, a DDoS assault generates more network streams than a spoofing assault, usually less (Neto et al., 2023).

Data preprocessing

The model’s performance is enhanced, and more accurate features are produced due to the critical data preparation phase. In this stage, categorical data is preprocessed into integer values using a robust scalar and label encoding technique.

Robust scalar: This study uses the data preparation method RobustScaler. Compared to other scaling techniques like the StandardScaler or MinMaxScaler, which are more susceptible to outliers, the RobustScaler is specifically created to scale the characteristics of a dataset. The RobustScaler scales the data utilizing the median and Interquartile range (IQR), as opposed to the mean and standard deviation used by the StandardScaler. The median is less dependent on outliers than the mean, while the IQR is a measure of the data’s dispersion that is likewise less affected by outliers.

Label encoder: Label encoding renders the input numerical labels into a machine-learning algorithm (Sharma et al., 2020).

Data splitting: After performing the data preprocessing steps, the models are built. The data is divided into training and testing data with a ratio of 80% training data and 20% testing data to enhance accuracy and efficacy for this phase. After splitting, the model is trained using the different variants of the deep learning model.

Deep neural network

This study inspected DNN’s aptitude for spotting cyberattacks in IoT settings. According to Deng & Yu (2014), the deep learning method incorporates the learning area that employs irregular data in numerous stages through organizational frameworks. Deep learning synthesizes graphical design, neural networks, and pattern recognition. The deep learning model performs well for prediction on large data sets. According to Huang et al. (2013), DNNs are exceptional to other machine learning classifiers. The proposed deep learning technique examines the organic patterns on a sample of network stream packets.

Additionally, deep learning logically supports multi-task training, which uses a single-layer deep neural network to consider all building forms’ properties. This network mainly consists of self-learning units with two or more layers. DNN uses hidden units among the input and output layers. The hidden unit p can use a logistic function to translate the scalar variable yq of subsequent layers to the input xq below. In a DNN network, (4) and (5) can predict the output of ith neuron yi as following Eqs. (1) and (2):

(1) yi=f(ξi),

(2) f(ξi)=ϑ+Σhετi−1WiXj,

f(ξi) represents the transfer function and ξi is the potential of ith-neuron. The transfer function is shown in the following Eq. (3):

(3) f(ξi)=11+exp(−ξi)

The sum of squared errors can represent the entire objective cost function where the output neurons are used to determine the target values, yo, and yo^ as shown in Eq. (4).

(4) C=Σ1/2(yo−yo^)

Different variants of DNN models are utilized for this research. This study employs a sequential DNN1 algorithm with a single input layer. The first input layer has 256 units and employs the relu activation method. The hidden layer comprises two dense and one dropout layer. The dense layer comprises 256 and 34 units with relu and softmax activation functions, and the dropout layer has 256 units. The output layer is the next activation function, relu and softmax, used to tackle the categorical categorization obstacle.

For the next DNN variant model, this study uses the sequential DNN2 model. The input layer is 1 with 256 units and employs the same method as DNN1. The hidden layer comprises two dense and two dropout layers. The dropout layers comprised 256 and 128 units. The dense layer contains 128 and 34 units with relu and softmax activation functions, respectively. This research uses the sequential DNN3 algorithm with one input layer as a third DNN variant model. The input layer is 1 with 256 units and employs a similar method as DNN1. The hidden layer is next, which comprises three dense and three dropout layers. The dropout layers comprised 256, 128 and 64 units. The dense layer contains 128, 64 and 34 units with relu and softmax activation functions, respectively. The DNN model has utilized Adam as an optimizer to compute and decrease the damage utilizing sparse_categorical_crossentropy.

Convolutional neural network

Convolutional neural network(CNN) is a subclass of deep neural networks that have demonstrated exceptional performance in several computer vision applications. This research uses the CNN model for text classification. The CNN model comprises three consequential layers: a max-pooling layer, a convolutional layer, and a fully connected layer. Different variants of CNN models are utilized for this research. This study uses a sequential CNN1 model with a single input layer. The model has 10 layers: one convolutional layer, three Leaky_ReLU layers, one max-pooling layer, one dropout layer, three dense layers, and one flattened layer. The convolutional block is merged with a 1D convolutional neural network, one Leaky_ReLU layer, one max-pooling layer, and a dropout layer with a 30% dropout rate. The consequent attribute maps tend to be flattened after the convolutional and pooling layers. Following flattening, the attributes proceed through two Leaky_ReLU layers, three dense layers with 64, 32, and 34 units using the softmax function.

This research uses the sequential CNN2 model as a second CNN variant model. The model has 14 layers: two convolutional layers, four Leaky_ReLU layers, two max-pooling layers, two dropout layers, three dense layers, and one flattened layer. The convolutional block is merged with a 1D convolutional neural network, Leaky_ReLU layer, max-pooling layer, and a dropout layer with a 30% dropout rate. After the convolutional and pooling layers, subsequent attribute maps often become flat. Following flattening, the attributes proceed through two Leaky_ReLU layers, three dense layers with 64, 32, and 34 units using the softmax function.

This study uses the sequential CNN3 model for the next CNN variant model. The model has 18 layers: three convolutional layers, five Leaky_ReLU layers, three max-pooling layers, three dropout layers, three dense layers, and one flattened layer. The convolutional block is merged with a 1D CNN, Leaky_ReLU layer, max-pooling layer, and a dropout layer with a 30% dropout rate. The consequent attribute maps tend to be flattened after the convolutional and pooling layers. Following flattening, the attributes proceed through two Leaky_ReLU layers, three dense layers with 64, 32, and 34 units using the softmax function. The output layer employed the Adam optimizer in the final section to compute and reduce the loss employing categorical_crossentropy.

Recurrent neural network

A recurrent neural network (RNN) is a genre of neural network architecture constructed to work with data sequences. RNNs retain a hidden state that gathers data from earlier time steps in the sequence, unlike standard feedforward neural networks, which analyze each data point independently (Alzubaidi et al., 2021).

Various variants of RNN models are utilized for this research. This study uses a sequential RNN1 system with a single input layer. The shape of the input layer is 1 with 32 units, and the relu activation function is used. The dropout regularisation is used with the input layer. Next, add the output layers comprising one flattened layer, three dense layers and two Leaky_ReLU layers. The dense layer with 34, 16 and 32 utilizes the activation functions softmax and a fully connected layer.

The sequential RNN2 algorithm that utilizes a single input layer is the following RNN variant. The input layer used the same method and function as RNN1 variants. The dropout regularisation is used with the input layer. Next, add the second RNN layer with 32 units and use the dropout regularisation layer. The output layers comprise one flattened layer, three dense layers and two Leaky_ReLU layers. The dense layer with 34, 16 and 32 utilizes the activation functions softmax and a fully connected layer.

This study uses the sequential RNN3 model with one input layer as a third DNN variant model. The input layer used the same method and function as RNN1 variants. The output layers comprise one flattened layer, three dense layers and two Leaky_ReLU layers. The dense layer with 32, 16, and 34 uses the softmax activation functions and is a fully connected layer. The output layer employed the Adam optimizer in the final section to compute and reduce the loss employing categorical_crossentropy.

Algorithm 1 illustrates the method of the proposed model for cyberattack detection. A proposed method for anticipating cyberattacks in an IoT environment using deep learning techniques is described in the presented methodology. The algorithm proceeds by specifying the input as cyberattack data and the intended output as predictors of cyberattacks. We propose a function called CICIoTDataset that performs data preprocessing operations on the data, such as employing the RobustScaler for scaling features and label encoding for categorical labels to a numerical form. The processed data is returned with the symbols x (features) and y (labels). The function known as TrainModel is defined. To do so, the data are divided into training and testing sets ( xtrain,xtest,ytrain,ytest). Next, three different kinds of NN models are built: a DNN, a CNN, and a RNN. It then returns the created model. The DNN and RNN model used the hyperparameter keras.backend.clear_session() and hidden initializer random_uniform(SEED). It is explained how DDoSPrediction works. An iterative process involves a specified number of epochs E_p and batch sizes B_s. The model is applied to generate predictions (x) from the input data Data for every epoch and batch size. The crossentropy loss is determined between the actual labels X and the expected results. After that, the algorithm outputs the expected outcomes.

Algorithm 1 Proposed algorithm for cyberattacks detection model.

1: Input: CICIoT Dataset	
2: Output: Cyberattack Prediction	
3: Dp = Data Preprocessing	
4: a) Robust Scalar	
5: b) Label Encoding	
6: (x, y) ←Dp	
7: Train Model (x, y)	
8:    Splitting = x_train,x_test,y_train,y_test	
9:    DNN = Sequential DNN()	
10:   CNN = Sequential CNN(), keras.backend.clear_session(), random_uniform(seed=SEED)	
11:   RNN = Sequential RNN(), keras.backend.clear_session(), random_uniform(seed=SEED)	
12: Return model	
13: DDoSPrediction (model)	
14: for every E_p epochs	
15:     for every B_s in the batch-size	
16:         x = model(Data);	
17:         Loss = cross_entropy, compute loss	
18: Return ← predicted result	

Results and discussion

We evaluate the effectiveness of the proposed model on a CICIoT2023 dataset. The evaluation metrics used are listed below to predict a proposed methodology.

Accuracy: Accuracy is an important assessment parameter used in performance evaluation because it is the ratio of effectively accurate predictions to all positive predictions made by the model. This value is proportionally illustrated by Eq. (5), which facilitates comprehension of the metric conceptual equation.

(5) Accuracy=TP+TNTP+TN+FP+FN.

Precision: The precision of a model or system shows the accuracy with which it predicts the positive class. It indicates the confidence level regarding the model’s capacity to generate positive predictions and conveys its accuracy. Equation (6) illustrates this value proportionally, making the metric conceptual equation easy to understand.

(6) Precision=TPTP+FP.

Recall: Recall, often called sensitivity, provides an assessment measure focusing on the ratio of all positive cases to the proportion of accurate positive predictions. This balanced perspective provides a unique advantage during estimating, as Eq. (7) computation demonstrates. This formula illustrates the usefulness of recall as an accessible gauge for evaluating model performance.

(7) Recall=TPTN+FN

F1-score: The appropriately titled F1 score serves as a harmonic mean of memory and precision since it may efficiently convey the essence of balanced performance. Combining these two measures results in the F1-score, a widely used estimate of model performance that is particularly helpful in assessment. Equation (8), which appears complex but offers much insight, accurately describes this fundamental estimation computation.

(8) F1−measure=2×Precision+RecallPrecision+Recall

Deep neural network

The results of a study on the performance of different deep neural network (DNN) models are presented in Table 2. Several significant performance measures were used in the evaluation to determine these models’ effectiveness in classification tasks. The table thoroughly summarizes the findings, facilitating comprehension of each model’s performance on 33 cyberattacks. Model DNN1 achieved 87.42% accuracy, 86.94% precision, 87.42% recall, and 86.26% F1-score. Model DNN2 demonstrated 84.73% accuracy, 85.69% precision, 84.73% recall, and 84.39% F1-score. Model DNN3 exhibited 88.64% accuracy, 91.20% precision, 88.64% recall, and an F1-score of 88.51%. Concerning accurate classification, handling false positives and negatives, and achieving a balance between accuracy and recall, these measures collectively provide insights into the effectiveness of each DNN model. In contrast to the other models, Model DNN2 performs poorly in accuracy, with Model DNN3 emerging as the best model in precision and F1-score.

Table 2 Result of deep neural network.

Models	Accuracy	Precision	Recall	F1-score	
DNN1	87.42	86.94	87.42	86.26	
DNN2	84.73	85.69	84.73	84.39	
DNN3	88.64	91.20	88.64	88.51	

Figure 2 visualizes the results of the DNN1 model. Figure 2A graph illustrates the accuracy for both training and validation. The training accuracy was 0.676% at the 0th epoch and ranged among drops and boosts until roughly 0.851% at the end. Validation accuracy starts at 0.775% at the 0th epoch and varies among losses and gains until it comes near 0.875% at the last epoch. Figure 2B loss curve shows the training and validation loss. training loss started at 0.799% and fell to 0.01% at the last epoch. Validation loss started at 0.0299% at the 0th epoch and fell to 0.01% by the 30th epoch.

Figure 2 Visualization of DNN1 model.

Figure 3 demonstrates the results of the DNN2 model. Figure 3A graph displays the training and validation accuracy. the training accuracy was 0.651%, and it varied between drops and gains until about 0.848% at the 0th epoch. Validation accuracy ranges among losses and gains until it peaks at 0.848% at the last epoch. Figure 3B graph displays the training and validation loss. Training loss decreased to 0.10, and validation loss remained 0.001%.

Figure 3 Graphical representation of DNN2 model.

Figure 4 represents the outcomes of the DNN3 model. Figure 4A graph represents the training and validation accuracy. The training accuracy peaked at about 0.87% accuracy at the last epoch. Validation accuracy peaked at 0.875% accuracy at the last epoch. Figure 4B presents the training and validation loss. Training loss decreased to 0.01 at the last epoch. Validation loss remains the same 0.01% at the last epoch.

Figure 4 Graphical visualization of DNN3 model.

Convolutional neural network

Table 3 demonstrates the performance results of various convolutional neural network (CNN) models in a classification assessment of 33 cyberattacks in an IoT environment. Experiments have been conducted on CNN1, CNN2, and CNN3 models. The proportion of effectively classified occurrences among all instances is how accurately a classification is made. CNN3 exhibited the highest accuracy, scoring 96.37%, ahead of CNN2 (94.30%) and CNN1 (95.49%). Precision measures the percentage of precise positive predictions of favorable outcomes. CNN3 had the best precision, scoring 96.15%, while CNN2 and CNN1 followed in at 94.74% and 95.17%, respectively. A recall measures the proportion of real positive instances correctly anticipated as positive. Recall for CNN3 was 96.37%, CNN1 was 95.49%, and CNN2 was 94.30%. The F1-score is a balanced indicator of a model’s performance because it is the harmonic mean of precision and recall. CNN1 had an F1-score of 94.48 percent, CNN3 had a 95.51%, and CNN2 had 93.36%. Based on these outcomes and the offered assessment metrics, the CNN3 model performs best for cyberattack detection.

Table 3 Result of convolutional neural network.

Models	Accuracy	Precision	Recall	F1-score	
CNN3	96.37	96.15	96.37	95.51	
CNN2	94.30	94.74	94.30	93.36	
CNN1	95.49	95.17	95.49	94.48	

Figure 5 demonstrates the results of the CNN1 model. Figure 5A shows the training and validation accuracy graph. The training accuracy was 0.68% at the 0th epoch and fluctuated between falls and increases until roughly 0.94% at the 30thepoch. Validation accuracy starts at 0.76% at the 0th epoch and varies between gains and losses until it comes near 0.95% at the 30th epoch. The graph in Figure 5B represents the training and validation loss. Training loss is about 0.01 at the last epoch. Validation loss decreased to 0.01% at the last epoch.

Figure 5 Graphical visualization of CNN1 model.

Figure 6 represents the outcomes of the CNN2 model. Figure 6A depicts the training and validation accuracy graph. Training accuracy rises to 0.94%, and validation accuracy is 0.73%. The training and validation loss is represented on the graph in Fig. 6b. Training loss decreases to 0.01, and validation loss decreases to 0.01%.

Figure 6 Graphical representation of CNN2 model.

Figure 7 demonstrates the results of the CNN3 model. The training and validation accuracy is represented by the graph in Fig. 7a. The training accuracy is 0.90%; following several cycles of gains and losses, it goes up about 0.955%. Validation accuracy is 0.94%. The training and validation loss is depicted on the graph in Fig. 7b. Training loss decreased to 0.13, and validation loss peaked at about 0.10%.

Figure 7 Graphical visualization of CNN3 model.

Recurrent neural network

The comparison analysis of three RNN models, RNN1, RNN2, and RNN3, is shown in Table 4. The RNN1 model had a 96.52% accuracy rate. It had a high level of precision (96.25%), showing that a substantial amount of the positive predictions were accurate. A sizable percentage of effective captures of positive experiences is indicated by the recall value (96.52%). The F1-score (95.73%) indicates an adequate balance between recall and precision, letting the model perform well. A 96.01% accuracy was attained using the RNN2 model. A good capacity for making accurate positive predictions is indicated by the precision number (95.77%). This model detected many positive cases, with a recall value of 96.00%. The F1-score (95.65%) indicates a balanced trade-off between recall and precision, which adds to its strong performance. The RNN3 model had a 95.89% accuracy rate. The accuracy (95.60%) shows a noteworthy capacity for making precise optimistic predictions. The model has successfully detected several positive instances, as indicated by the recall value (95.89%). The F1-score (95.03%) shows that precision and recall are often well-balanced.

Table 4 Result of recurrent neural network.

Models	Accuracy	Precision	Recall	F1-score	
RNN1	96.52	96.25	96.52	95.73	
RNN2	96.00	95.77	96.00	95.65	
RNN3	95.89	95.60	95.89	95.03	

Figure 8 represents the results of the RNN1 model. Figure 8A graph represents the training and validation accuracy. Training accuracy is about 0.60%, and validation accuracy is 0.75%. The training and validation loss is depicted on the graph in Fig. 8b. Training loss seems to decrease to 0.01, and validation loss to 0.01%.

Figure 8 Graphical visualization of RNN1 model.

Figure 9 demonstrates the results of the RNN2 model. The training and validation accuracy is represented by the curve in Fig. 9a. The training accuracy reaches about 0.94% accuracy at the last epoch. Validation accuracy increases to 0.95% accuracy at the last epoch. The training and validation loss is depicted on the graph in Fig. 9b. Training loss started from 0.24% at 0th epoch and declined to 0.10, and validation loss reached about 0.10%.

Figure 9 Graphical visualization of RNN2 model.

The RNN3 model is visualized in Fig. 10. Figure 10A graph represents the training and validation accuracy. The training accuracy is about 0.95% accuracy at the last epoch. Validation accuracy is 0.71% at the start and increases at 0.95% accuracy at the last epoch. Figure 10B shows the training and validation loss. Training loss decreased to 0.10, and validation loss declined to 0.10%.

Figure 10 Graphical visualization of RNN3 model.

Comparison with existing study

Table 5 presents a comparison with the benchmark study (Neto et al., 2023). It can be seen that Neto et al. (2023) have an accuracy of 99.43%, a precision of 70.54%, a recall of 91.05%, and an F-score of 71.92%, whereas the proposed approach attained an accuracy of 96.52%, a precision of 96.25%, a recall of 96.52%, and an F-score of 95.73%. In this context, the Proposed Approach outperforms the base article regarding accuracy, precision, recall, and F-score, suggesting our approach is better for cyberattack detection in a realistic IoT environment.

Table 5 Comparison of the proposed approach with benchmark study.

Ref.	Precision	Recall	F-score	
Neto et al. (2023)	70.44	83.15	71.40	
Proposed approach	96.52	96.52	95.73	

Findings and discussion

IoT has become widely used in various applications due to the rapid progress of interconnected technology. Concurrently, the threat of cyberattacks is becoming more difficult to resolve. The experiment is performed with multiple DL variants using the CICIoT2023 to resolve this issue. Accuracy, precision, recall, and F1-measure optimization metrics are used to assess the model’s efficacy. The effectiveness of deep learning models, their capacity for generalization, and their significance are assessed using statistical analysis. “Model complexity” indicates the degree of intricacy and sophistication of a DL model’s structure and its capacity to find patterns and connections in the data. A model becomes more complex when additional parameters are included. A network’s properties increase with its number of neurons and layers. While parameter diversity adds complexity to the processing load, it also helps DL models recognize complex patterns in the data. This is often mitigated by normalization techniques such as batch normalization, weight deterioration, and dropout, which increase the complexity of the model. Numerous methods reduce model complexity by regularising the loss function and adding consequence terms. Discouraging extremely complex parameter values helps prevent overfitting. Multiple deep learning variants (DNNs, RNNs and CNNs) have increased the prediction performance. This study uses deep learning variants to address the cyberattacks in the IoT environment. The experiment results show that the proposed deep learning variants perform more accurately and efficiently than conventional techniques. The test findings show that the proposed approach is more successful than other cyberattack detection algorithms.

Conclusion and future scope

This study suggests that a deep learning model, which is more efficient compared to a machine learning model, can be used to categorize cyberattacks in an IoT environment. The DNNs, CNNs, and RNNs model was selected as a workable contender for this study because it incorporates feature extraction and selection into its model, making it preferable to crude machine learning techniques. In the current study, DNNs, CNNs, and RNNs were utilized to categorize threats and attacks using the CICIoT2023 dataset. Compared to traditional models, the RNN model, which is utilized as a deep learning model, has a high accuracy rate for cyberattack categorization of roughly 96.5%. Additionally, utilizing RNN and graph neural networks to identify cyberattacks guides future work on IoT malware detection. In the future, incremental training will be included by observing network streams. To upgrade the device with an innovative method of assault. Additionally, we will investigate whether it is possible to integrate our proposed model into real-time cyber-attack detecting systems.

Supplemental Information

Supplemental Information 1 Code for IoT Attack detection.

Click here for additional data file.

Additional Information and Declarations

Competing Interests

Author Contributions

Data Availability

The authors declare that they have no competing interests.

Sidra Abbas conceived and designed the experiments, performed the experiments, analyzed the data, performed the computation work, prepared figures and/or tables, authored or reviewed drafts of the article, and approved the final draft.

Imen Bouazzi conceived and designed the experiments, performed the experiments, analyzed the data, performed the computation work, authored or reviewed drafts of the article, and approved the final draft.

Stephen Ojo conceived and designed the experiments, performed the experiments, performed the computation work, authored or reviewed drafts of the article, and approved the final draft.

Abdullah Al Hejaili conceived and designed the experiments, analyzed the data, performed the computation work, authored or reviewed drafts of the article, and approved the final draft.

Gabriel Avelino Sampedro conceived and designed the experiments, performed the experiments, analyzed the data, performed the computation work, prepared figures and/or tables, authored or reviewed drafts of the article, and approved the final draft.

Ahmad Almadhor conceived and designed the experiments, performed the experiments, analyzed the data, prepared figures and/or tables, authored or reviewed drafts of the article, and approved the final draft.

Michal Gregus conceived and designed the experiments, prepared figures and/or tables, authored or reviewed drafts of the article, and approved the final draft.

The following information was supplied regarding data availability:

The CIC IoT Dataset 2023 is available at https://www.unb.ca/cic/datasets/iotdataset-2023.html.

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
