# Peer review of "Evaluating deep learning variants for cyber-attacks detection and multi-class classification in IoT networks"

_PeerJ Computer Science, doi:10.7717/peerj-cs.1793_

## Round 0.1 · original submission · Major Revisions

I have received reviews of your manuscript from scholars who are experts on the cited topic. They find the topic very interesting; however, several concerns must be addressed regarding experimental results, contributions, and comparisons with current approaches. These issues require a major revision. Please refer to the reviewers’ comments listed at the end of this letter, and you will see that they are advising that you revise your manuscript. If you are prepared to undertake the work required, I would be pleased to reconsider my decision. Please submit a list of changes or a rebuttal against each point that is being raised when you submit your revised manuscript.

Thank you for considering PeerJ Computer Science for the publication of your research.

With kind regards,

**Language Note:** The review process has identified that the English language must be improved. PeerJ can provide language editing services - please contact us at [email protected] for pricing (be sure to provide your manuscript number and title). Alternatively, you should make your own arrangements to improve the language quality and provide details in your response letter. – PeerJ Staff

Reviewer 1 ·

Basic reporting

The basic reporting of the manuscript is clear. The English language used is clear and the manuscript is quite easy to follow. The number of references used in the study is not enough. Mainly, Research gaps are missing in this manuscript. There is no analysis of the limitations of any of the previous work. In most cases, there is not even a summary, just a name-check, and reference. It is suggested to add greater depth to the analysis in the Related Work section and refer to recent articles.

Experimental design

• The experimental setup seems good. The proposed method is not compared with the state-of-the-art/related works in terms of experimentation,results, and suitability.

Validity of the findings

• It looks like some significant findings were observed. However, the findings can be evaluated after clarifying the models.

Additional comments

In my view, the manuscript requires a major revision before it can be published in a journal.

Reviewer 2 ·

Basic reporting

The innovation and contribution of the research are not clear.

Experimental design

no comment.

Validity of the findings

no comment

Additional comments

The innovation of the research should be mentioned clearly in the abstract section.
The grammatical and typo errors should be corrected.
The evaluation must be performed using more datasets.
The performance of the proposed method should be compared with more related techniques.
What are the limitations of the existing methods? How these limitations are solved by the proposed method?
All the variables in the equations must be defined.
More recent related works must be reviewed.
What are the limitations and future work of the proposed method.

---

## Round 0.2 · accepted · Accept

I am pleased to inform you that your work has now been accepted for publication in PeerJ Computer Science.

Please be advised that you are not permitted to add or remove authors or references post-acceptance, regardless of the reviewers' request(s).

Thank you for submitting your work to this journal. On behalf of the Editors of PeerJ Computer Science, we look forward to your continued contributions to the Journal.

With kind regards,

Reviewer 1 ·

Basic reporting

The manuscript has been revised as per the reviewer's suggestions and it can be considered for publication.

Experimental design

-

Validity of the findings

-

Additional comments

-

Reviewer 2 ·

Basic reporting

no comment

Experimental design

no comment

Validity of the findings

no comment

Additional comments

The paper is well revised. I accept this paper